# Dyeing Properties, Color Gamut, and Color Evaluation of Cotton Fabrics Dyed with *Phellodendron amurense* Rupr. (Amur Cork Tree Bark)

**DOI:** 10.3390/molecules28052220

**Published:** 2023-02-27

**Authors:** Xinyu Ji, Zhijun Zhao, Yulu Ren, Fei Xu, Jianhong Liu

**Affiliations:** 1Department of Clothing and Costume Design, College of Art & Design, Qiqihar University, Qiqihar 161006, China; 2Department of Mechanical and Electrical Engineering, Qiqihar School of Engineering, Qiqihar 161005, China

**Keywords:** *Phellodendron amurense* Rupr., natural dye, dyeing, cotton fabric, color gamut space, color evaluation

## Abstract

The application of plant dyes in the textile industry has been very limited due to their limited sources, incomplete color space, and narrow color gamut, etc. Therefore, studies of the color properties and color gamut of natural dyes and the corresponding dyeing processes are essential for completing the color space of natural dyes and their application. In this study, water extract from the bark of *Phellodendron amurense* (*P. amurense*) was used as a dye. Dyeing properties, color gamut, and color evaluation of dyed cotton fabrics were studied, and optimal dyeing conditions were obtained. The results showed that the optimal dyeing process was pre-mordanting with liquor ratio at 1:50, *P. amurense* dye concentration at 5.2 g/L, mordant concentration (aluminum potassium sulfate) at 5 g/L, dyeing temperature at 70 °C, dyeing time of 30 min, mordanting time of 15 min, and pH 5. Through the optimization of the dyeing process, a maximum color gamut range was obtained with lightness *L** value from 74.33 to 91.23, *a** value from −0.89 to 2.96, *b** value from 4.62 to 34.08, chroma *C** value from 5.49 to 34.09, and hue angle *h*° value from 57.35° to 91.57°. Colors from light yellow to dark yellow were obtained, among which 12 colors were identified according to the Pantone Matching Systems. The color fastness against soap-washing, rubbing, and sunlight on the dyed cotton fabrics all reached grade 3 level or above, further expanding the applicability of natural dyes.

## 1. Introduction

Since the birth of synthetic dyes, they have rapidly become the main dyeing materials for fibers and other materials. Synthetic dyes have developed rapidly due to their advantages in their simple production process, low cost, complete color space, vivid colors, high color fastness, and amenability to standardization and industrialization, etc. [1,2]. However, toxic substances such as heavy metals that remain in the textiles are harmful to the human body, and the wastewater discharged from the textile industry poses a serious threat to the ecosystem [3,4]. Countries around the world have put great importance and have made regulations on this matter. The development of ecological dyes has become the focus of research in the textile field [5]. Among them, natural dyes have attracted the attention of consumers because of their natural ecological and biodegradable properties and their therapeutic functionalities [6,7,8]. In addition, the unique aesthetic value and added value of natural dyes [9,10] could be marketing tools, providing clothing industries and brands with a modern and fashionable aspect [11]. However, the application and industrialization of natural dyes have many limitations, which include low color repeatability, a narrow color range, a lack of dyeing standardization, and poor fastness [12,13,14,15].

Among all the natural fibers, cotton is the main raw material in the textile industry due to its economic value and popularity [16]. Natural dyes have been widely used for dyeing natural fibers. The study of dyeing properties of natural dyes on cotton fabrics has become an important research direction. However, most of the plant dyes show low affinity with cotton fabrics. Cotton cellulose structure contains large amount of hydroxyl groups that become hydrolyzed and ionized with negative charges in water. Most plant dyes are anionic dyes. They are also negatively charged in water-based solutions because of their functional groups, such as hydroxyl and carboxyl groups. The negative charges of dye molecules and cotton fibers repel each other, resulting in a low color fastness and narrow color space [17]. To solve these problems, research has focused on the modification of cotton fabrics, the optimization of metal mordants [18,19], and the use of bio-mordants. For most plant dyes, dyeing cotton fabrics requires the modification of cotton fabrics. Cotton fabrics modified by chitosan [20,21] and protein [22,23] have been shown to have improved dyeing color characteristics and color fastness. However, additional modification processes increase dyeing costs and reduce dyeing efficiency.

*Phellodendron amurense* (*P. amurense*) is a species of tree in the family *Rutaceae*, commonly called Amur cork tree, with the Chinese name huáng bò. *P. amurense* is mainly located in northeast and north China, the far east of Russia, southern Sakhalin, Japan, and Korea. The main chemical compositions of the inner layer of *P. amurense* bark are alkaloids, lactones, phenolic acids, and phenylpropanoids, etc. [24]. Among them, several alkaloids [25], such as isoquinoline alkaloids [26], are the main components of *P. amurense* dye, including berberine (C_20_H_18_NO_4_), palmatine (C_21_H_22_NO_4_), and jatrorrhizine (C_20_H_20_NO_4_), in quantity-descendant order [27]. Their chemical structures are shown in Figure 1. Research on *P. amurense* has mainly focused on their various pharmaceutical effects, and it has been widely applied in clinical treatment [28,29,30]. In recent years, the application of *P. amurense* dye in the textile industry has also made great progress. Many natural fabrics such as silk, wool, cotton, and some synthetic fabrics can be dyed directly with *P. amurense* [31]. Using aluminum, iron, or copper metal salt mordant can improve the dyeing effect and the color fastness [32]. Fabrics dyed with *P. amurense* have certain antibacterial, anti-UV, anti-oxidative-fading, and other effects [33]. Among natural dyes, *P. amurense* belongs to the rare category of natural cationic dyes. Without modification, cotton fabrics directly dyed with *P. amurense* showed excellent dyeing properties [34]. More studies have focused on the dyeing performance of *P. amurense* and the pharmaceutical effects of dyed fabrics. There is a lack of systematic research on the color characteristics, color gamut, and color value of natural fibers dyed with *P. amurense*, which limits the application and industrialization of the dye.

In this study, taking color characteristic values of the dyed fabrics as the evaluation standards, the optimum dyeing parameters of *P. amurense* on cotton fabrics were obtained through single-factor experiments. The effect of dyeing parameters on widening the color gamut was discussed. The color characteristics, color gamut, and color value of cotton fabrics dyed with *P. amurense* were further evaluated.

**Figure 1 molecules-28-02220-f001:**
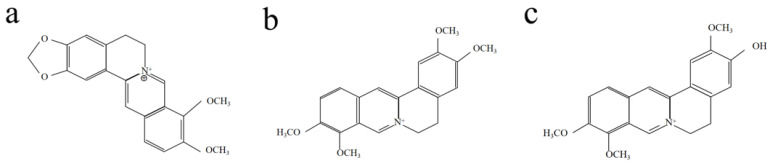
Chemical structure of the main dye compounds in *P. amurense*: (**a**) berberine, (**b**) palmatine, (**c**) jatrorrhizine.

## 2. Results and Discussion

### 2.1. UV-Visible Spectroscopy of Phellodendron amurense Dye Solution

The UV-visible spectroscopy result of the *P. amurense* dye solution is shown below in Figure 2. Three absorption peaks were found. Among them, the absorption peaks at 280 nm and 328 nm in the UV range were typical of berberine, palmatine, and jatrorrhizine, which are isoquinoline alkaloids with an isoquinoline ring and a substituted benzene ring [35]. The absorption peak at 418 nm was the absorption of the dye within the visible light range.

### 2.2. Effect of Mordanting on the Color Characteristics of the Cotton Fabrics

In the CIE-Lab color space system (according to ISO 105-J01:1997), *L**, *a**, and *b** are the three axes. *L** represents the perceptual lightness from black to white [0, 100]. *a** and *b** represent the four colors: red, green, blue, and yellow. *a** represents the green–red opponent colors within a range of [−128, 127]; *b** represents the blue–yellow opponent colors within a range of [−128, 127]. *C** is the purity value, indicating the color from dull to bright within a range of [0, 100]. *h°* is the hue angle within a range of [0, 360], which gives the information of the color type (e.g., 0°—red, 60°—yellow, 120°—green, 180°—cyan, 240°—blue, 300°—magenta, and 360°—red). The *K*/*S* value represents the light absorption of the dyed fabric at its maximum absorption wavelength. The higher the *K*/*S* value is, the darker the color is.

The color characteristic values of dyed fabrics from dyeing process No. 1 to 4 in Table 1 are shown in Figure 3a. It can be seen that the *b** value of all the dyed fabrics ranged from 9.15 to 35.06, which represents the yellow color. Among them, the apparent color of the dyeing and pre-mordanting fabrics is relatively close, both showing a bright yellow tone. The mordanting process showed an impact on the quality of the yellow tone, which can be seen from the *b** value of different dyeing processes: pre-mordanting (35.06) > dyeing (27.24) > meta-mordanting (24.15) > post-mordanting (9.15). The dyeing effect of pre-mordanting was significantly enhanced, which can also be seen from the *K*/*S* value, as in Figure 3b.

Without mordant, the interaction between the anions of cotton fabric fibers and the cations of the *P. amurense* dye could help the adsorption of the dye molecules onto cotton fibers. This was in fact a physical adsorption of the hydroxyl groups in the cotton fibers and the hydroxyl groups in the dye molecules through hydrogen bonds, as shown in Figure 3c. The bond energy of hydrogen bonds is between 25 to 40 kJ/mol [36], and their stability is far weaker than that of covalent bonds and ionic bonds. Natural dyes therefore have limited affinity with cotton fibers. After adding KAl(SO_4_)_2_·12H_2_O mordant, the aluminum ion Al^3+^ could form a coordination complex with the dye molecules and cotton fibers, which increased the dye uptake rate and the color fastness. The main components of *P. amurense* dye are berberine, palmatine, and jatrorrhizine, all of which have methoxyl -OCH_3_ group. -OCH_3_ is an electron-donating group. Since the electron-donating conjugation effect is greater than the electron-withdrawing induction effect, the overall structure of the dye molecules is electron-donating. In the mordanting process, the Al^3+^ ion and the hydroxymethyl group of the cotton fabrics and the methoxyl group of the *P. amurense* dye undergo a coordination reaction to form Al-O bonds, as shown in Figure 3d. The bond energy of Al-O is 512 kJ/mol [36], which can effectively improve the binding force between the dye and the fiber molecules and improve the dyeing efficiency and color fastness of dyed fabrics.

The *b** values of the *P. amurense* dyed cottons fabrics show that the dyeing colors obtained from all the dyeing processes were mainly yellow. The optimum dyeing process of *P. amurense* on cotton fabrics was pre-mordanting.

**Figure 3 molecules-28-02220-f003:**
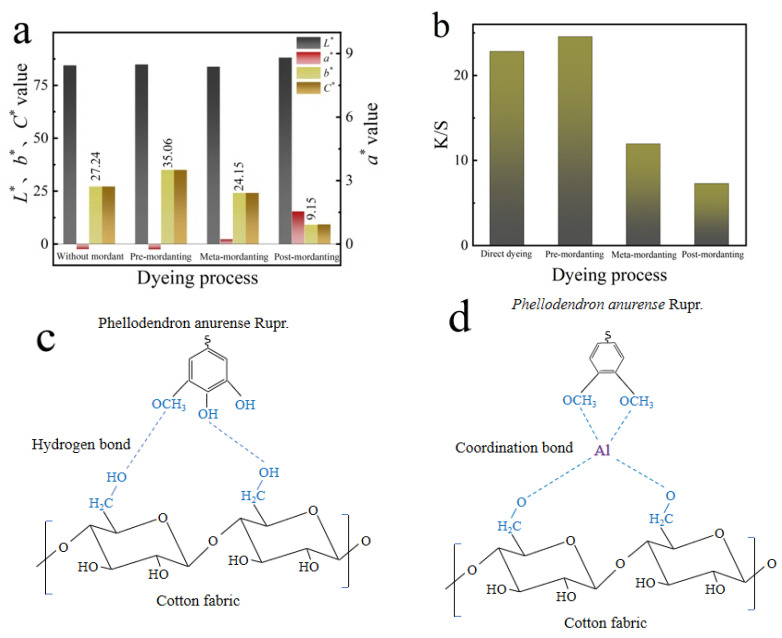
(**a**) Effect of mordanting on the color characteristics of dyed cotton fabrics; (**b**) effect of mordanting on the *K*/*S* value of dyed cotton fabrics; (**c**) interaction of *P. amurense* dye with cotton fabrics; (**d**) interaction of *P. amurense* dye with Al^3+^ and cotton fabrics.

### 2.3. Effect of Dyeing Parameters on the Color Characteristics of Dyed Cotton Fabrics

Based on the pre-mordanting process, single-factor experiments (No. 5 to 10 in Table 1) were carried out, and the color characteristic values of *P. amurense* dyed cotton fabrics were measured (Figure 4). The optimum dyeing parameters were obtained.

As shown in Figure 4a, color characteristic values of cotton fabrics increased with the increase in dyeing temperature. When the dyeing temperature reached 70 °C, the *b** value reached its maximum (34.76), and the lightness of yellow color was the highest. As the dyeing temperature increased, the adsorption of dye on cotton fabrics reached a balance. High temperature favors increase in the swelling effect of cotton fibers, which helps the penetration of dye molecules into the fibers. High temperature also favors the coordination of dye molecules, Al^3+^ ions, and cotton fibers. However, excessively high temperatures have a negative effect on the adsorption [37], causing the dye to desorb from the fiber, resulting in a decrease in lightness of the yellow color. The optimum dyeing temperature in the pre-mordanting process was 70 °C.

As shown in Figure 4b, lightness *L** value decreased gradually and reached a balance, while *a**, *b**, and *C** values increased gradually and reached a balance with the increase in dyeing time. The cotton fabrics’ adsorption capacity for dyes increased rapidly within a short period of time. When the dyeing time reached 30 min, the *b** value has reached its maximum (34.63), and the adsorption of dyes on the fabrics has saturated. The optimum dyeing time in the pre-mordanting dyeing process was 30 min.

As shown in Figure 4c, the mordant concentration had a slightly greater impact on the lightness *L** value than on other values. As the mordant concentration increased, the *L** value gradually decreased to 84.51; the *b** value decreased, but quickly reached 35.28. During the mordanting process, excess aluminum ions can combine with dye molecules and cause aggregation, making it difficult for dye molecules to diffuse into the interior of fabric fibers, resulting in a slow dye uptake. In addition, the apparent color of the dyed cotton fabrics showed a darker color because of excess mordant, and the color purity was reduced. Increases in mordant concentration would also increase the cost of wastewater treatment [38]. The optimum mordant concentration in the pre-mordanting dyeing process was 5 g/L.

As shown in Figure 4d, the *L** and *b** values were mostly affected by the pH value of the dye solution. As the pH value increased, the *L** value gradually decreased, while the *b** and *C** value gradually increased and then decreased. When pH was lower than 5, the dye solution was bright yellow, and some precipitates were formed. The *L** value was higher, while the *a**, *b**, and *C** values were lower. The dyeing effect was low. At pH 5, the dye solution has gradually changed from bright yellow to yellowish brown, and the precipitates have gradually disappeared. At this point, the *b** and *C** values were the highest (35.37 for both). The apparent color of the fabrics was bright yellow, and the dyeing effect was the best. When pH was higher than 5, the dye solution became maroon, and there were no precipitates. The *L**, *b**, and *C** values decreased. The dyeing effect was not good. Alkaloids have good solubility in weak acidic solution. While *P. amurense* dye molecules have good stability under weak acidic conditions, under strong acidic or alkaline conditions, the dye molecules are not stable and can undergo structural change. Especially under strong acidic conditions, they aggregate to form larger particles and are difficult to diffuse into the fibers, resulting in a low dye uptake rate [39]. The pH value of the dye solution had a great influence on the stability of the *P. amurense* dye. The optimum pH in the pre-mordanting process was 5.

In summary, the optimum condition of dyeing cotton fabrics with *P. amurense* was pre-mordanting with dyeing temperature at 70 °C, a dyeing time of 30 min, a mordant concentration of 5 g/L, and a dyeing pH of 5. Cotton fabrics dyed from *P. amurense* under the optimum conditions were further studied for their color gamut range by modifying the solution concentration and number of dyeing times.

As shown in Figure 4e, with the increase in dye concentration, the characteristic values of color and *b** and *C** values showed an obvious increase, which represented a gradual increase in yellow light and higher purity. Other color characteristic values changed little. When the dye concentration was 5.2 g/L, the dyed fabrics showed the maximum *b** and *C** value. A high solution concentration can increase the concentration gradient of the dye molecules between the exterior and the interior of the fiber, making it easier for dye molecules to enter the cotton fibers. Therefore, changing the *P. amurense* dye concentration could be used to obtain a yellow color that ranges from light to dark.

As shown in Figure 4f, with the increase in dyeing times, the *L**, *b**, and *C** values decreased gradually; the *a** value varied from −0.89 to 0.99, which showed a slight transition from green to red light. When the number of dyeing times reached nine, all the color features became stable. During the dyeing process, when the same solution and residual solution were used for dyeing, the dye molecules in the residual solution gradually decreased. After a certain number of dyeing times, it is difficult for more dye molecules to adsorb to the fiber surface, making it difficult to increase the dyeing rate. The number of dyeing times impacted the *L** value and *b** value the most. The optimal number of dyeing times was nine.

**Figure 4 molecules-28-02220-f004:**
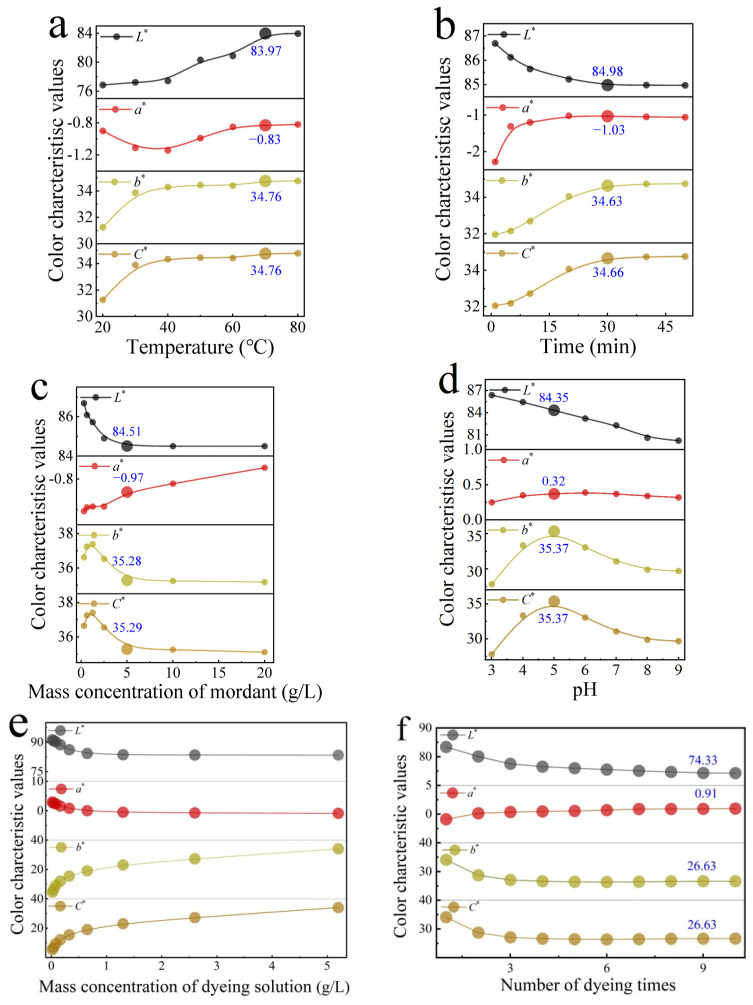
Effect of dyeing parameters of pre-mordanting process on the color characteristic values. (**a**) Dyeing temperature, (**b**) dyeing time, (**c**) mordant concentration, (**d**) pH of the dye solution, (**e**) mass concentration of dye solution, (**f**) number of dyeing times.

### 2.4. Color Coordinates and Color Gamut Range

In the *P. amurense* pre-mordanting process, cotton fabrics dyed with different dye concentrations and number of dyeing times (dyeing process No.9 to 10 in Table 1) were used to analyze the color characteristics and color gamut range. Color samples were analyzed in the two-dimensional CIE-Lab system with the *a** value as the horizontal axis and the *b** value as the vertical axis, as in Figure 5a. Colors samples were then analyzed in a two-dimensional coordinate system with the concentration of the dye solution as the horizontal axis and the lightness *L** value and purity *C** value as the vertical axis, as shown in Figure 5b. All color samples were analyzed in a hue angle *h*° range from 0° to 180°, as shown in Figure 5c.

As can be seen in Figure 5a, changing the *P. amurense* dye concentration could obtain colors from light to deep yellow with *a** values from −0.89 to 2.96 and *b** values from 4.62 to 34.08. Repeating the dyeing process could obtain deeper yellow tones with *a** values from −0.89 to 0.99 and *b** values from 26.63 to 34.08. As shown in Figure 5b, with the increase in the *P. amurense* dye concentration, the lightness *L** value decreased from 91.23 to 83.40, and the purity *C** value increased from 5.49 to 34.09. Repeating the dyeing process could obtain lightness *L** values from 74.33 to 83.40 and purity *C** values from 26.35 to 34.09. From Figure 5c, changing the concentration of the *P. amurense* dye solution could obtain colors with hue angles *h°* from 57.35° to 91.57°. Repeating the dyeing process could obtain hue angles *h°* from 88.08° to 91.57°.

The pre-mordanting of cotton fabrics with *P. amurense* dye could obtain colors with an overall range of lightness *L** values from 74.33 to 91.23, *a** values from −0.89 to 2.96, *b** values from 4.62 to 34.08, and purity *C** values from 5.49 to 34.09. The overall range of the hue angles *h°* was from 57.35° to 91.57°.

**Figure 5 molecules-28-02220-f005:**
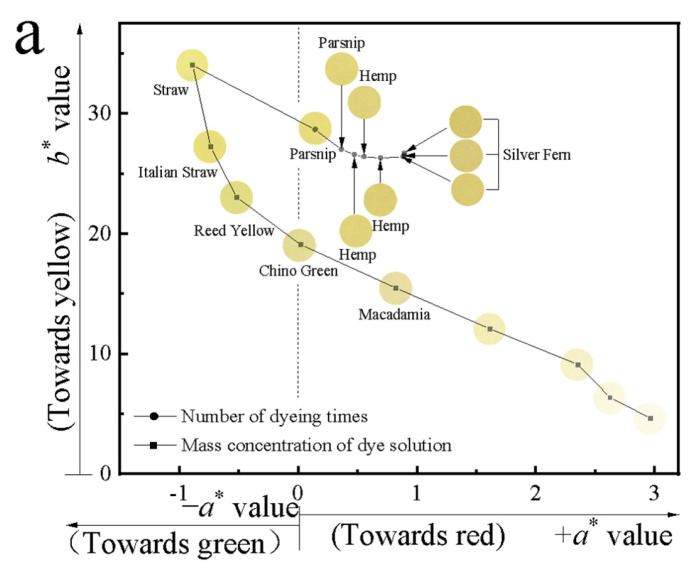
Color range of cotton fabrics dyed with *P. amurense* in pre-mordanting processes. (**a**) *a** and *b** range, (**b)** lightness *L** range and purity *C** range, (**c**) the hue angle *h*° range.

### 2.5. Color Fastness and Color Evaluation of the Cotton Fabrics

Color fastness, including the soap-washing fastness, rubbing fastness, and light fastness of dyed cotton fabrics from the dyeing process without mordant and aluminum potassium sulfate pre-mordanting process were shown in Table 2. In the dyeing process without mordant, none of the color fastnesses of the fabrics reached the standard level, with soap-washing fastness between levels 2 and 3 and light fastness between levels 1 and 2. In the pre-mordanting processes, however, soap-washing fastness, rubbing fastness, and light fastness all reached level 3 and above.

The disadvantage of alkaloid dyes is that the color fastness to light of the dyed fabrics is very poor, at only level 1 to 2. The cationic group of *P. amurense* dye is relatively sensitive, and it is in a conjugated ring system. Upon contact with light energy, the positive charge is more likely to be activated from the position of the cationic group and transferred to the entire ring system, causing the bond to be destroyed and the color to fade [40]. By adding aluminum potassium sulfate mordant, Al^3+^ acted as a central ion and combined with the cotton fabrics and dye molecules to form a complex, which effectively improved the combination force between the dye and the fibers and increased the dyeing effect and color fastness of the dyed fabrics—especially the sunlight fastness, which met textile standards.

According to the color difference tolerance standard (Δ*E_ab_* ≤ 3.0) specified in GB/T 31430-2015, the dyed fabric samples were firstly screened. Color numerical comparison was made between 13 samples and the Pantone Matching Systems (ColorTell© 2023), as shown in Figure 5a. Among them, 12 colors with 0 < Δ*E* < 3 were identified within the Pantone Matching System. One sample color with Δ*E* > 3 was not in the Pantone system. The Pantone Matching System is an international standard. Thus, the colors that were obtained from *P. amurense* dyed cotton fabrics show good application potential and economic value.

## 3. Materials and Methods

### 3.1. Materials and Chemicals

Cotton fabrics (bleached, 100% cotton, density 162.5 g/m^2^, warp density 358/10 cm, weft density 200/10 cm, thickness 0.89 mm) were sourced from Chunyan Garment factory, Rizhao economic development zone, Shandong, China. *P. amurense* was purchased from Anguo Tongyi Traditional Chinese Medicine Yinpian Co., Ltd., Hebei, China. Aluminum potassium sulfate dodecahydrate KAl(SO_4_)_2_·12H_2_O, citric acid (C_6_H_8_O_7_), and sodium hydroxide (NaOH) were purchased from Bkman Biotechnology, Hebei, China. All chemicals used were analytical grade. Soap (standard) was purchased from Soap Factory Shanghai (Shanghai, China). Distilled water was made in the laboratory.

### 3.2. Methods

#### 3.2.1. Cotton Fabrics Pre-Treatment

Cotton fabrics were cut into pieces (8 cm × 4 cm) and soaked in NaOH solution for 30 min under room temperature. They were then rinsed and dried for future use. Before the dyeing process, cotton fabrics were soaked in distilled water for 10 min. After removing excess water, the wet cotton fabrics were ready to use.

#### 3.2.2. Dye Extraction

*P. amurense* bark was washed thoroughly with distilled water and dried in the drying oven to constant weight. Dried bark was cut into smaller pieces and ground for 1 min (39,000 r/min) in a grinder. The powder was then passed through a 22-mesh sieve.

A mass of 100 g *P. amurense* powder was mixed with 1 L distilled water and soaked for 24 h. The mixture was then heated to 100 °C for 2 h for dye extraction. The concentrated solution was further filtrated and freeze dried for 12 h (Yetuo YTLG-10 Desktop lyophilizer, Shanghai Yetuo Technology, Shanghai, China); 5.2 g of extract was obtained. The *P. amurense* dye solution that was used for future dyeing process was made by adding 5.2 g dye powder into 1 L distilled water, for a concentration of 5.2 g/L.

#### 3.2.3. Dyeing Process

Single-factor experiments were carried out to test different dyeing processes. Dyeing parameters in different dyeing processes are shown in Table 1. Process No. 1 to 4 were to test the effect of mordanting on the color of dyed fabrics. Process No. 5 to 9 were to test the effect of parameters in the pre-mordanting process on the color of dyed fabrics. Process No. 10 was to test the effect of repeat dyeing on the color of dyed fabrics.

The liquor ratio in all the dyeing processes was 1:50. KAl(SO_4_)_2_·12H_2_O was used as mordant. The dyeing and mordanting temperature were both 70 °C. Cotton fabrics were firstly put in a dyebath of 40 °C, and the dyebath temperature was then increased to the target temperature at a speed of 1.5 °C/min. The mordanting time was 15 min. Citric acid and NaOH were used to adjust the pH of the dye solutions. Fabrics were rinsed after dyeing and dried under room temperature.(a)Dyeing (without mordant): cotton fabrics were dyed directly in the *P. amurense* dye solution.(b)Mordanting:Pre-mordanting: cotton fabrics were treated firstly in aluminum potassium sulfate solution for mordanting. The mordanted fabrics were then wringed out and dyed in the dye solution.Meta-mordanting: Cotton fabrics were dyed in the dyebath containing both dye and the mordant.Post-mordanting: Cotton fabrics were dyed firstly in the dye solution. They were then wringed out and treated within aluminum potassium sulfate mordant solution.(c)Repeat dyeing: Cotton fabrics were treated firstly in aluminum potassium sulfate solution for mordanting. The dyeing process was then applied and repeated.

#### 3.2.4. UV-Visible Spectroscopy

A 5.2 g/L *P. amurense* dye solution was made by mixing the dye powder in distilled water. The solution was then diluted 100 times to measure its absorption spectrum under a range of wavelengths (190 nm–800 nm) using a UV-visible spectrometer (Spotlight 400, PerkinElmer, Waltham, MA, USA).

#### 3.2.5. Color Characteristic Measurement

Dyed cotton fabrics were equilibrated for more than 24 h at room temperature before the measurement. The *K*/*S* value and color characteristic values (CIE-Lab color space system) were measured using Desktop spectrophotometer (Color Eye^®^ 7000A, X-Rite, Grand Rapids, MI, USA) by employing CIE standard illuminant D_65_, with a measuring geometry of d/8° and 10° as the view angle. Fabrics were folded into 4 layers, on which 3 points were randomly chosen and measured, and the average value was taken.

#### 3.2.6. Color Fastness Experiments

Color fastness experiments of all the samples were carried out according to the ISO standards. Washing fastness and rubbing fastness tests were carried according to ISO 105-C10:2006 and ISO 105-X12:2001. Greyscale grades evaluated the results of wash fastness from 1 to 5. Light fastness was carried according to ISO 105-B02:1994/Amd.2:2000. Blue-scale grades evaluated the lightfastness property with values from 1 to 8.

## 4. Conclusions

As a relatively rare natural cationic dye, *P. amurense* dye has a good affinity to non-modified cotton fabrics and can dye and obtain satisfactory colors. It has been proved that bright yellow colors can be obtained without adding mordant or adding aluminum salt mordant. The pre-mordanting process with aluminum salt can effectively improve the light fastness of dyed cotton fabrics, which met the fastness standards required for the commercialization of textiles, which is very important and necessary for natural yellow dyes. The raw materials used, the dyeing process, and the color fastness of *P. amurense* dyeing all meet the NODS (Natural Organic Dye Standard) [41].

In addition, using color characteristic values as the evaluation standards to evaluate the dyeing performance and color gamut of cotton fabrics dyed with *P. amurense* dye is a relatively intuitive, simple, and effective method. Changing the mass concentration of *P. amurense* dye and the number of dyeing times under the optimal dyeing process conditions can effectively increase the color gamut and modify the color tone from light yellow to dark yellow. The characterization method and color evaluation studies in this paper are useful for the in-depth development of natural yellow dyes and evaluating the color value of natural dyes. They can also provide new research ideas and methods for solving the problem of the narrow color space of natural yellow dyes.

## Figures and Tables

**Figure 2 molecules-28-02220-f002:**
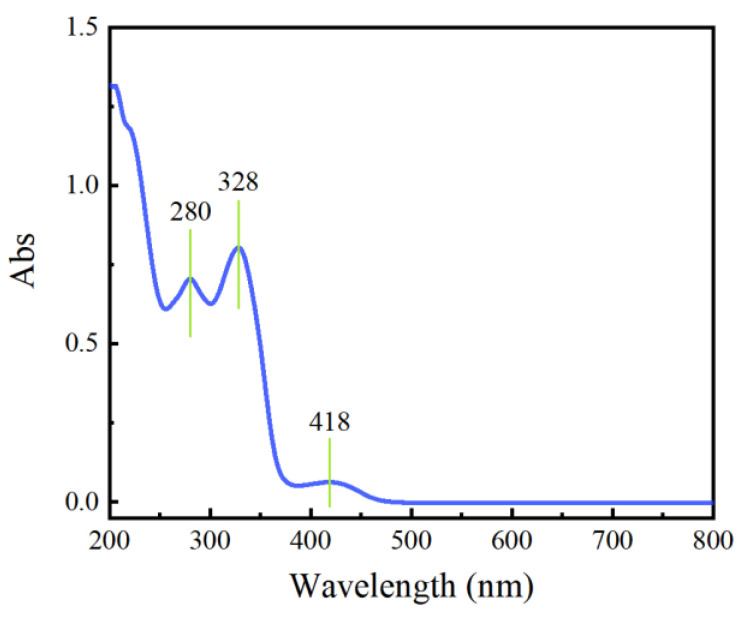
Absorption spectrum of *P. amurense* dye solution.

**Table 1 molecules-28-02220-t001:** Color fastness and color evaluation of dyed cotton fabrics.

Dye Process	Mass Concentration of Dye Solution (g/L)	Number of Dyeing Times	Soap Washing Fastness		Rubbing Fastness		Light Fastness	PANTONE (TPX)		Compare with PANTONE (Δ*E*)
			Color Change	Color Staining	Dry	Wet		Color Number	Color Name	
Without mordant	5.20	1	2–3	2–3	3	3	1–2	13-0917	Italian Straw	0.976
Pre-mordanting	0.33	1	3	3	4	3	3	12-0709	Macadamia	3.647
	0.65	1	3	3	4	3	3	13-0613	Chino Green	1.514
	1.30	1	3	3	4	3	3	13-0915	Reed Yellow	0.425
	2.60	1	3	3	4	3	3	13-0917	Italian Straw	1.556
	5.20	1	3	3	4	3	3	13-0922	Straw	1.718
	5.20	2	3	3	4	3	3	14-0925	Parsnip	1.818
	5.20	3	3	3	4	3	3			2.263
	5.20	4	3	3	4	3	3	14-0721	Hemp	2.711
	5.20	5	3	3	4	3	3			2.744
	5.20	6	3	3	4	3	3			2.824
	5.20	7	3	3	4	3	3	15-0719	Silver Fern	2.524
	5.20	8	3	3	4	3	3			2.448
	5.20	9	3	3	4	3	3			2.375

**Table 2 molecules-28-02220-t002:** Parameters in different dyeing processes.

No.	Dyeing Process	Mass Concentration of Dye Solution (g/L)	Dyeing Temperature (°C)	Dyeing Time (min)	Mordant Concentration (g/L)	pH	Number of Dyeing Times
1	Dyeing	5.2	70	30	——	5	1
2	Pre-mordanting	5.2	70	30	5	5	1
3	Meta-mordanting	5.2	70	30	5	5	1
4	Post-mordanting	5.2	70	30	5	5	1
5	Pre-mordanting	5.2	20~80	30	5	5	1
6	Pre-mordanting	5.2	70	1~60	5	5	1
7	Pre-mordanting	5.2	70	30	0.31~20	5	1
8	Pre-mordanting	5.2	70	30	5	3~9	1
9	Pre-mordanting	0.02~5.2	70	30	5	5	1
10	Pre-mordanting	5.2	70	30	5	5	2~10

—— is used to indicate absence.

## Data Availability

The data presented in this work are available in the article.

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
