# Peer review of "Dyeing Properties, Color Gamut, and Color Evaluation of Cotton Fabrics Dyed with Phellodendron amurense Rupr. (Amur Cork Tree Bark)"

_molecules, 2023, doi:10.3390/molecules28052220_

Round 1

Reviewer 1 Report

Dear authors,

The data provided here for the article entitled "Dyeing Properties, Color Gamut and Color Evaluation of Cotton Fabrics dyed with Phellodendron amurense Rupr. (Amur Cork Tree Bark)" are interesting; however, I am offering some comments throughout the manuscript.

(1) Line 17: What do you mean by dyeing solution?

(2) Line 20: This is not correct (hue a* value …, hue b* value).

(3) All short forms are not abbreviated. It is recommended to use abbreviations first and then continue in a short form.

(4) The introduction part is unprofessional; there is a lack of consistency between lines and paragraphs; it really needs to be revised very carefully.

(5) The main research gap of the work is totally absent in the Introduction. The last paragraph of the Introduction should provide information (only) about the science gap in the previous studies and what motivates you to do this review with the objective of the study.

(6) The authors are recommended to cite recent articles.

(7) Line 90: Specifications of cotton fabric such as fabric type (woven or knitted), yarn densities should be mentioned.

(8) Line 107: The word ''ground'' is correct.

(9) Lines 111-114: Authors have stated that ''And the dye powder was obtained for future dyeing process. The dyeing solution that will be used for the dyeing process was decided and calculated as the ratio of the mass of dye powder obtained per litre of liquid used for extraction'' but the powder producing method is not mentioned. How do you measure the ratio of the mass of dye powder obtained per litre of liquid?

(10) Lines 116-139: The dying and mordanting method must be clear and understandable.

(11) Line 127: dyeing process should be pre-mordant dyeing process.

(12) Lines 154-15: Authors stated that color fastness experiments of all the samples were carried out according to the ISO standards but they mentioned Chinese Standard ID. Why?

(13) As the fastness properties of the dyed fabrics are limited, the authors might discuss a possible segment for the technical application of the dye from Phellodendron amurense.

(14) Color fastness is low even after mordanting. You must take help from some cationizing agents.

(15) The application of mordant in a pre-dyeing step modifies the colour of the cotton substrate a lot. Authors should report colour change due to mordanting expressed as K/S mordanted prior to dyeing.

(16) The authors are recommended to mention the ecological issue due to the use of metals in dyeing.

(17) Results must be explained with proper reason. 

(18) The entire conclusion must be re-written with conclusive findings and by retaining coherence.

(19) I am not an English speaker, but I found many typos and grammatical errors throughout the manuscript. These must be corrected and revised.

(20) References should be in accordance with the journal template.

Author Response

Thank you for your comprehensive and quick review and correction, which makes our article further possible. Your comments and corresponding modifications are attached below for your review.

Reviewer 2 Report

Reviewer Comments

molecules-2179977

Title: Dyeing Properties, Color Gamut and Color Evaluation of Cotton Fabrics dyed with Phellodendron amurense Rupr. (Amur Cork Tree Bark)

1.      There is lack of literature in the some natural dyeing articles containing berberin, palmatine and jatrorrhizine. The below subjects of articles will complete this gap and thus improve its quality:
 (e.g. Organic cotton fabric dyed with dyer's oak and barberry dye by microwave irradiation and conventional methods¸ DOI: 10.35530/IT.072.01.1755;

Examination of Dyeing Properties of the Dyed Cotton Fabrics with Barberry (Berberis vulgaris L.), https://www.tandfonline.com/doi/full/10.1080/15440478.2018.1558143; ect.).

2.      In section 2.2.1, is the cotton fabric applied to the bleached fabric or to the unbleached fabric? This should be clearly stated. If bleaching has been done, a bleaching process should also be added.

3.      Freeze dried process should be clearly stated.

4.      Figure 2. Absorption spectrum of Phellodendron anurense was given.  Phellodendron anurense includes three main coloring compounds Berberin, Palmatine and Jatrorrhizine. The spectrum contains total coloring compound (berberin, palmatine and jatrorrhizine). Spectra can be given separately (as berberin, palmatine and jatrorrhizine) instead of the total coloring compounds. For analysis, I can recommend HPLC or LC-MS instead of UV-visible spectroscopy.

5.      In the conclusion, it should be compared with NODS (Natural Organic Dye Standard) and strongly emphasized that the study complies with NODS (Natural Organic Dye Standard; https://www.tandfonline.com/doi/pdf/10.1080/15440478.2022.2162187?needAccess=true&role=button).

The manuscript is comprehensive natural dye study. It will contribute to natural dyeing and textile sector. It can be published after the specified deficiencies are corrected.

Reviewer 3 Report

-          What is the difference between dyeing duration and dyeing time

-          You conclude that the effect dyeing time was 30 min while you list the time in the table was 1

-          How you have dyed cotton fabric at pH 2? it is well known at this pH the cotton gose to dissolve and tensile was destroyed. In addition, suitable pH for dyeing cotton fabric at alkaline pH and you didn’t examine it

         How you reach pH 2 using citric acid

Reviewer 4 Report

The paper is well written and a lot of work has been done however what is the novelty of this work?

If I look through the reference list of the paper or do a simple internet search I find a number of pieces of work on cotton dyeing using Phellodendron Amurense as a dye. This includes colour analysis of the dye. To be published a piece of work should be novel. Please show how this work is different and what is the novel science that it is adding to the field. 

I find the graphs almost impossible to understand and would advise revising them.

Fig 3a multiple formats are given to show L* a* and b* values in this work. Please keep a consistent format. Is the truncating on the L* axis needed?

Fig 4 These graphs are not easy to understand. Please consider a different way to give this data. Possibly one graph for L* showing all of the different mordant variations and the same for a* and b*. In their current form they are too small to compare or derive valuable information from. Dye concentration of 4e has been truncated but there are no units in lower truncation to provide where it was truncated.

The a* b* graph of fig 5a is good however it would be useful to have a vertical figure next to it showing the corresponding change in L*

Fig 5a what does "number of dyeing" and "mass concentration of dye solution" refer to? How is the Pantone colour name relevant to scientific work?

Fig 5b What does "concentration of dyeing solution" and "dyeing times" refer to? What is the number in some of the markers? What are the pink numbers on the graph? what are the grey and green markers?

Is fig 5c needed as it replicates the results of fig 5a?

Reviewer 5 Report

Dear authors,

The manuscript is very well prepared and structured. The topic could be of interest to a wide range of researchers, especially those involved in dyeing with natural dyes. The discussion is thorough and well written and is supported by the results obtained. It is suggested that some facts in the manuscript be corrected and improved as follows:

- page 3, line 127: it is suggested to replace the term dyeing process with the term Pre-mordanting dyeing process

- page 4, lines from 145 to 152: the measurement geometry is missing (Is it d/8°?)

- page 4, lines 153-157: The standard for the evaluation of the colour change is missing.

- page 5, lines from 172 to 173: Please add a reference to where it was written that the values of CIE a* and CIE b* are between -128  and 127.

- page 5, line 194 and line 204: Please add a reference to the energy of the H-bond and the Al-O-bond.

- nowhere is it written that a blue scale with values from 1 to 8 was used for the evaluation of light fastness.

Yours sincerely,

The reviewer of the manuscript

Round 2

Reviewer 1 Report

I have no further observations.